# A Systemic and Integrated Analysis of p63-Driven Regulatory Networks in Mouse Oral Squamous Cell Carcinoma

**DOI:** 10.3390/cancers15020446

**Published:** 2023-01-10

**Authors:** Alexandra Ruth Glathar, Akinsola Oyelakin, Kasturi Bala Nayak, Jennifer Sosa, Rose-Anne Romano, Satrajit Sinha

**Affiliations:** 1Department of Biochemistry, Jacobs School of Medicine and Biomedical Sciences, State University of New York at Buffalo, Buffalo, NY 14203, USA; 2Department of Oral Biology, School of Dental Medicine, State University of New York at Buffalo, Buffalo, NY 14214, USA

**Keywords:** p63, COTL1, OSCC, HNSCC, metastasis, biomarkers

## Abstract

**Simple Summary:**

Oral squamous cell carcinomas (OSCC) are the most common malignancies affecting the oral cavity and account for 40% of all head and neck squamous cell carcinoma cases. Unfortunately, patient outcomes are generally unfavorable, as diagnosis normally occurs at a late stage of the disease, as well as a lack of effective targeted treatments. Our identification of potential prognostic biomarkers is of particular importance to the field and could lead to the generation of effective targeted therapies which may improve patient outcomes and survival. Here, we utilize a mouse model of OSCC to explore the role of p63 as an important oncogenic transcription factor in the control of OSCC proliferation and migration. We also generate a p63-driven gene expression signature for mouse OSCC which identifies both novel and conserved genes and pathways, which may be relevant in human disease and which may serve as potential biomarkers and targets for future therapeutics.

**Abstract:**

Oral squamous cell carcinoma (OSCC) is the most common malignancy of the oral cavity and is linked to tobacco exposure, alcohol consumption, and human papillomavirus infection. Despite therapeutic advances, a lack of molecular understanding of disease etiology, and delayed diagnoses continue to negatively affect survival. The identification of oncogenic drivers and prognostic biomarkers by leveraging bulk and single-cell RNA-sequencing datasets of OSCC can lead to more targeted therapies and improved patient outcomes. However, the generation, analysis, and continued utilization of additional genetic and genomic tools are warranted. Tobacco-induced OSCC can be modeled in mice via 4-nitroquinoline 1-oxide (4NQO), which generates a spectrum of neoplastic lesions mimicking human OSCC and upregulates the oncogenic master transcription factor p63. Here, we molecularly characterized established mouse 4NQO treatment-derived OSCC cell lines and utilized RNA and chromatin immunoprecipitation-sequencing to uncover the global p63 gene regulatory and signaling network. We integrated our p63 datasets with published bulk and single-cell RNA-sequencing of mouse 4NQO-treated tongue and esophageal tumors, respectively, to generate a p63-driven gene signature that sheds new light on the role of p63 in murine OSCC. Our analyses reveal known and novel players, such as COTL1, that are regulated by p63 and influence various oncogenic processes, including metastasis. The identification of new sets of potential biomarkers and pathways, some of which are functionally conserved in human OSCC and can prognosticate patient survival, offers new avenues for future mechanistic studies.

## 1. Introduction

Oral squamous cell carcinomas (OSCCs) are the most common malignancies arising from the mucous membrane of the oral cavity and typically affect the buccal and labial mucosae, gingiva, floor of the mouth, hard palate, lips, tongue, and cheeks [1,2,3]. OSCC is linked to tobacco exposure, alcohol consumption, and human papillomavirus infection and accounts for 40% of all head and neck squamous cell carcinoma (HNSCC) cases [1,2,3]. The outcomes of patients with OSCC are generally unfavorable because diagnosis typically occurs at a late stage of the disease and there are no effective treatments [1,2,3]. To this end, recent genomic and transcriptomic characterizations of human OSCC/HNSCC tumors have offered crucial molecular insights into the disease’s etiology and progression. [4,5,6]. Harnessing the power of such data-rich resources to identify biomarkers for targeted therapy, however, remains a challenge and necessitates parallel studies of OSCC, particularly in well-defined genetic models

In mice, 4-nitroquinoline 1-oxide (4NQO)-induced carcinogenesis produces a spectrum of preneoplastic and neoplastic lesions that mimic the progression, molecular, and histological changes of OSCC in humans [7,8]. Administration of 4NQO in drinking water or by topical application induces sequential changes in murine oral epithelium, from hyperplasia and dysplasia to the development of in situ carcinomas [7,8]. These lesions exhibit up to 94% similarity with the mutational landscape of human carcinogen-induced HNSCC [9], highlighting the potential of this model for identifying human-relevant biomarkers. Several studies have used 4NQO-induced carcinogenesis to uncover the molecular drivers of OSCC [9,10,11,12,13]. Deletion of *p16^INK4a^*, a gene in one of the most commonly mutated genomic loci in human OSCC, has been found to upregulate the expression of the oncogenic transcription factor *Trp63* (p63) and enhance the development of 4NQO-induced tumors [12]. p63 is a master regulator of epithelial development, shown in many mouse and human studies to drive oncogenesis [12,13,14,15,16,17,18,19]. The human gene *TP63* encodes two major isoforms, ΔNp63 and TAp63, with ΔNp63α being the predominant isoform expressed in most cells of epithelial origin and epithelial cancers [20,21,22,23]. *TP63* is amplified and overexpressed in most OSCC tumors and promotes tumor growth by multiple mechanisms [12,13,14,15,16,17,18,19,20,21,22,23]. Interestingly, the downregulation of p63 in late-stage and metastatic tumors and a 4NQO model of HNSCC facilitates the metastatic migration of cancer cells to secondary sites [24,25]. Despite the overwhelming evidence of a strong link between p63 and various facets of epithelial cancers, the underlying molecular mechanisms and the full repertoire of the downstream target genes, by which p63 promotes and contributes to the development of OSCC, are not fully known.

Here, we have utilized a comprehensive repertoire of transcriptomic and genomic tools to uncover novel p63-driven oncogenic pathways and targets to better understand its role in murine OSCC (mOSCC). We perform the first comprehensive transcriptomic characterization using bulk RNA sequencing (RNA-seq) of 4NQO-induced mOSCC cell lines, highlighting the diversity of murine OSCC, as well as providing a robust dataset for future studies employing these cell lines. To investigate the molecular function of p63 in mOSCC and identify its transcriptional targets, we utilized both loss- and gain-of-function models of p63 to identify a well-defined p63-driven gene signature. We also show that our identified signature coincides well with signatures identified in recent publicly available bulk and single-cell RNA-seq of 4NQO-induced mOSCC tumors and mouse esophageal squamous cell carcinoma (ESCC) tumors, respectively. Results from our RNA-seq, Chromatin immunoprecipitation-sequencing (ChIP-seq), and phenotypic experiments with mOSCC cell lines demonstrate that p63 regulates a broad range of pathways and targets, which affects downstream pathways including cell adhesion, migration, and metastasis, as well as the novel target, *Cotl1*. Our findings reaffirm the notion that dynamic expression of p63, specifically the ΔNp63 isoform, regulates an extensive swath of transcriptional targets, as well as maintains broad oncogenic signaling. These gene regulatory networks are essential for a wide range of oncogenic processes in mOSCC and are also likely to be relevant in the human disease.

## 2. Materials and Methods

### 2.1. Cell Culture Studies

Cell lines B7E3, B4B8, and B7E11 were obtained from Dr. Carter Van Waes [26]. All cell lines were grown and maintained in high-glutamine DMEM with the following supplements: 10% FBS, 1% nonessential amino acids, and antibiotics. All cell lines were tested by the eMycoPlus Mycoplasma PCR detection kit (BulldogBio) to ensure they were bereft of any mycoplasma infection.

### 2.2. Generation of Knockdown and Overexpression of Cell Lines

Lentivirus-mediated depletion of p63 in B7E11 cells was performed using the pGIPZ system. GIPZ lentiviruses containing short hairpin RNAs (shRNAs) (clone IDs: V3LMM_508694 (sh1), V3LMM_523845 (sh2), and V3LMM_523849 (sh3)) targeting *Trp63* were generated with the help of the Gene Modulation Services Shared Core at Roswell Park Comprehensive Cancer Center. The pINDUCER20 vector comprising a neomycin resistance gene was used as previously described [27] for doxycycline-inducible expression of FLAG-tagged ΔNp63α in B7E3 cells. The FLAG-tagged ΔNp63α cDNA (a gift from Caterina Missero) was transferred to pInducer20 using recombination Gateway LR clonase Enzyme mix kit (Thermofisher, Waltham, MA, USA) and the resulting plasmid was confirmed by DNA sequencing. Viral infection and selection with either puromycin or neomycin were performed as described before [28].

### 2.3. Western Blot Analysis

Western blots were generated according to a previously described protocol [28]. Briefly, 5 µL protein lysates were loaded onto SDS-polyacrylamide gels and transferred to Immun-Blot polyvinylidene difluoride membranes (Bio-Rad Laboratories, Hercules, CA, USA). After blocking in 5% milk, the membranes were incubated first in primary antibodies against p63 (4A4, 1:20,000), COTL1 (Proteintech, 1:10,000), K14 (a gift from Dr. Rose-Anne Romano) [29], Vimentin (CST, 1:5000), MMP9 (Proteintech, 1:10,000), Fibronectin (SinoBiological, 1:5000), ITGB4 (Proteintech, 1:10,000), E-cadherin (CST, 1:5000), and K6 (a gift from Dr. Julie Segre), then with horseradish peroxidase-conjugated secondary antibodies corresponding to the host of the primary antibody, and then washed in Tris-buffered saline with 0.05% Tween-20. Protein expression was detected with the LumiGLO peroxidase chemiluminescent substrate kit (SeraCare, Milford, MA, USA), and membranes were imaged using a Bio-Rad ChemiDoc imaging system. Uncropped Western blot images can be found in Appendix A.

### 2.4. ChIP-seq of p63/ΔNp63

The iDeal ChIP-seq kit for transcription factors (C01010055; Diagenode, Denville, NJ, USA) and the associated protocols were used to perform ChIP-seq. B7E11 cells were grown to ~90% confluency and cross-linked in the supplied fixation buffer supplemented with 0.5% formaldehyde for 10 min. Lysates from the fixed cells were subsequently sonicated with a Diagenode Bioruptor to obtain sheared chromatin with an approximate fragment length of 150–400 bp. The ChIP experiments for p63 were carried out using 2 µg of pan-p63 4A4 antibody (Santa Cruz Biotechnology) and 2 µg of ΔNp63 antibody (E6Q3O; Cell Signaling Technology, Danvers, MA, USA). Libraries were prepared as described before [30]. ChIP DNA and input controls were then subjected to 50 bp single-end sequencing on an Illumina HiSeq 2500 instrument (Illumina Inc., San Diego, CA, USA), which resulted in 15–25 million reads per sample.

### 2.5. ChIP-seq Analysis

The raw ChIP-seq reads from all experiments were mapped to the *Mus musculus* genome (mm10) using Bowtie2 v2.3.4.1. Peak calling was then performed using MACS2 v2.1.0, with a minimum false-discovery rate cutoff of 0.05. Sequenced genomic input was used as the control for each experiment, and resultant peaks were matched to the nearest gene using GREAT analysis with default settings [31,32]. To visualize ChIP peaks, bam files were processed with deepTools v3.3.2 to generate bigwig files, which were then uploaded to IGV [33]. The R tool ChIPseeker was used to annotate p63 ChIP-seq peaks to the nearest genomic feature of the mm10 genome assembly [34]. Adobe Illustrator was used for final image processing.

### 2.6. RNA Isolation and Library Preparation for RNA-seq

Total RNA from cell lines was extracted using a Direct-zol RNA miniprep kit (Zymo Research, Irvine, CA, USA). The extracted RNA was snap-frozen on dry ice and stored at −80 °C until library preparation. For each RNA sample, cDNA libraries were prepared using the TrueSeq RNA sample preparation kit (Illumina). The libraries underwent 50 bp single-end sequencing on an Illumina HiSeq 2500 instrument. Quality control metrics were performed on raw sequencing reads using the FASTQC v0.11.9 application.

### 2.7. RNA-seq Analysis

Reads were mapped to the appropriate reference genome, GRCm38/mm10 build, with HISAT v2.1.0 [35]. Reads aligning to the reference genome were quantified with featureCounts v1.5.3 to generate a matrix of raw counts, which was then processed in R, a free program for statistical computing and graphics, to generate normalized expression values in transcripts per million, according to the method proposed by Wagner et al. [36]. Differential gene expression analysis comparing control to p63 knockdown was carried out using DESeq2 v1.24.0 [37]. Differentially expressed genes with a false-discovery rate of ≤0.1 were considered statistically significant.

### 2.8. qRT-PCR Analysis

Total RNA from B7E11 knockdown and B7E3 overexpression cell lines were extracted using the Direct-zol RNA miniprep kit. The RNA was reverse transcribed with the Bio-Rad iScript cDNA synthesis kit, according to the manufacturer’s instructions. The resulting cDNA was used for qPCR with Bio-Rad iQ SYBR green Supermix. A list of the qRT-PCR primers can be found in Appendix A.

### 2.9. HNSCC and mOSCC Dataset Analysis

RNA-seq data from patient samples were obtained from The Cancer Genome Atlas (TCGA). TCGA RNA-seq expression and survival datasets were downloaded from cBioPortal and the UCSC Xena Browser [38,39,40]. RNA-seq fastq files from mouse 4NQO-generated oral tumors were downloaded from GSE54246 [11] and realigned with HISAT v2.1.0 to the mm10 genome, and then quantified using featureCounts v1.5.3. Differential gene expression analysis was performed using DeSeq2 between the control and 4NQO-treated mouse tongues. Processed single-cell RNA-sequencing data for mouse esophageal squamous cell carcinoma generated by Wang et al. [41] were obtained from the GSA (CRA002118). Cells with less than 200 genes expressed and more than 10% mitochondrial genes expressed were filtered out to ensure that only high-quality cells were utilized in downstream analyses. Genes expressed in less than 3 cells were also filtered out of the analysis. After filtering, data were normalized and variable features were identified utilizing the selection method: “vst”. Cell clustering was performed using nearest neighbor construction and then clustered. Identified clusters were then annotated to identify known cell populations. For the analyses performed here, cells were filtered to retain only tumor cells, as identified through our annotation, and the distribution of p63 expression was evaluated to identify p63^low^ and p63^high^ clusters, and differential gene expression analysis was performed between the identified cell populations. Heatmaps for the scRNA-seq data were generated using the Seurat package [42].

### 2.10. Invasion Assay

B7E3 pIND-Trp63 cells were treated with 0, 50, or 200 ng of doxycycline (dox) for 24 h to induce p63 expression. Cells were plated at a density of 200,000 cells in the top of a Boyden chamber (Corning, product No. 354480) with a medium containing no serum and were allowed to invade and migrate for 12 h through a Matrigel matrix to the bottom chamber, which contained fetal bovine serum as an attractant. Cells were then fixed in formaldehyde and permeabilized with methanol according to a standard protocol. Cells on the bottom of the chamber were then stained with a mixture of Evan’s blue and methylene blue, and imaged with a Cytation 1 imaging plate reader (BioTek of Agilent, Santa Clara, CA); the stained area was calculated using ImageJ.

### 2.11. Spheroid Assay

B7E3 cells were grown in 96-well culture plates coated with 50% Matrigel (Corning, product No. 356237) for 6–8 days. To induce p63 expression, the cells were treated with 100 μg/mL doxycycline. The resulting spheroids were imaged using a 4× lens objective on a Cytation 1 imaging device; images were analyzed using ImageJ software. Graphing and statistical analysis were performed using Microsoft Excel (Microsoft Corp., Redmond, WA, USA). Student’s *t*-tests for samples with equal variance were used to determine statistical significance.

### 2.12. Motif Enrichment Analysis of Enhancers

The MEME Suite tool CentriMo was used for local motif enrichment analysis to determine which DNA-binding transcription factor motifs from the HOCOMOCO mouse (v11 FULL) database [43] were enriched within regions obtained from the p63 ChIP-seq in B7E11 cells. Motifs were ranked according to *p*-value.

### 2.13. Gene Ontology/Pathway Enrichment Analysis

The GREAT tool was used to annotate binding loci to the nearest gene [32]. Identified genes were then subjected to either the KEGG pathway analysis or WikiPathway analysis utilizing the DAVID functional annotation tool or the Broad Institute Gene Set Enrichment Analysis (GSEA) web tool [44,45,46,47,48].

### 2.14. Immunofluorescence and Immunohistochemistry

De-identified SCC patient samples were obtained from the archives of the Department of Oral and Maxillofacial Pathology, the University of New York at the University at Buffalo. An oral cavity tumor with a normal tissue microarray (US Biomax, Inc.—Tissue Array, T271b, Derwood, MD, USA) and SCC tissue slides were deparaffinized and sequentially rehydrated in decreasing concentrations of ethanol in water. After heat-induced antigen retrieval in sodium citrate, tissues were blocked and then incubated overnight with primary antibodies specific to p63 (4A4, 1:300), COTL1 (Proteintech, 1:200), or K14 (1:300), according to standard protocols. For immunofluorescence, the slides were incubated with appropriate fluorescent secondary antibodies, and coverslips were mounted with a mounting medium containing DAPI (Vector Laboratories, Newark, CA, USA) to stain nuclei, and then they were sealed for imaging. For immunohistochemistry, antibody labeling was visualized with an Impact DAB substrate kit (Vector Laboratories). Counterstaining was conducted using hematoxylin (Vector labs) before the slides were rinsed in tap water, air dried, and coverslipped with Permount mounting medium (Thermo Fisher Scientific, Waltham, MA, USA).

### 2.15. Statistics

Statistical analyses were performed using R. A Shapiro–Wilk test was performed to check the normality of data, and then either Student’s *t*-test or a Wilcoxon signed-rank test was performed according to whether the data were normally distributed. A *p*-value lower than 0.05 was considered significant.

## 3. Results

### 3.1. The Global Transcriptome of 4NQO-Generated mOSCC Cell Lines

To perform molecular studies of p63, we chose three tumorigenic murine oral SCC cell lines derived from BALB/c oral keratinocytes exposed to 4NQO [26]. These cell lines, B7E3, B4B8, and B7E11, showed low (undetectable), medium, and high p63 expression, respectively (Figure 1A), thus presenting excellent models to examine the molecular function of p63 in mOSCC. These cell lines are known to differ in their invasive abilities and the associated expression of epithelial–mesenchymal transition (EMT)-associated genes [49]. Thus, we first chose to characterize these cell lines by exploring the protein expression of EMT-associated factors through Western blotting. B7E3 and B7E11 cells had the highest and lowest levels of vimentin, respectively (Figure 1A), consistent with previous immunofluorescence findings [49]. B7E11 cells also had higher keratin levels than B4B8 and B7E3 cells (Figure 1A). B7E11 cells displayed high expression of epithelial EMT markers, whereas B7E3 cells showed high levels of mesenchymal-associated markers (Figure 1A). Interestingly, B4B8 cells appeared to display a hybrid EMT phenotype, with high levels of both epithelial- and mesenchymal-associated markers (Figure 1A). This partial EMT state is supported by previous findings showing that B4B8 cells migrate more than both B7E3 and B7E11 cells and also produce larger tumors than B7E11 cells [49].

First, to explore the variability in gene expression between these cell lines, we performed dimension reduction using principal component analysis (PCA) using the top 2500 most variable genes. PCA revealed that the most variation in gene expression was between B7E3 and B7E11 cells, with B4B8 cells lying in between, matching our Western blot results (Appendix A). Next, we performed RNA-seq of each cell line in order to explore their transcriptional landscape. Our differential gene expression analysis identified 9171 differentially expressed genes (DEGs) between B7E3 and B7E11 cells, and 8693 DEGs between B7E3 and B4B8 cells (Appendix A). To integrate the gene expression differences across the three cell lines and to simplify the downstream analysis, we identified the genes showing consistent upregulation and downregulation in the low p63-expressing (p63^low^) and high p63-expressing (p63^high^) cell lines. This combined analysis resulted in 2588 genes enriched in B7E3 cells and 2376 genes enriched in both B4B8 and B7E11 cells (Figure 1B, Appendix A). GSEA of the top 500 enriched genes from this analysis found that p63^low^ B7E3 cells were enriched in genes associated with inflammatory response, whereas the p63^high^ cells showed enrichment in EMT, hypoxia, and xenobiotic metabolism genes (Figure 1C,D). Together, these results revealed that the B4B8 and B7E11 cells share a similar gene expression profile that is distinct from that of the B7E3 mOSCC cells. Interestingly, this shared B4B8/B7E11 gene expression profile may be associated with the transcription factor p63, as evidenced by the enrichment of pathways shown to be regulated by p63, including the p53 pathway, Wnt beta-catenin signaling, and EMT [50].

### 3.2. Generating a p63 Cistrome in the 4NQO-Transformed B7E11 mOSCC Cell Line

To further characterize the p63–regulated oncogenic program in mOSCC cells, we next sought to identify target genes, whose regulatory regions are bound by p63. To this end, we performed independent ChIP-seq experiments in the p63^high^ B7E11 cells utilizing two p63-specific antibodies, 4A4 and ΔNp63 E6Q30 (Figure 1A). The 4A4 antibody recognizes all p63 isoforms and identified 29,001 genomic sites, whereas the ΔNp63-specific E6Q3O antibody identified 6414 sites (Figure 2A, Appendix A). Next, we used the intersectBed function in bedtools to identify the 6396 high-confidence p63-bound sites common to both antibodies (Appendix A). As expected, a motif analysis performed on these high-confidence peaks revealed that the consensus p63 motif (*p* = 1.4 × 10^−3985^) was the most enriched, followed closely by the p73 (*p* = 2.5 × 10^−3815^) and the p53 (*p* = 4.8 × 10^−2832^) motifs (Figure 2B). The distribution of p63 peaks relative to the transcriptional start site indicates that p63 preferentially targets distal intergenic and intronic regions, which are likely to act as enhancers (Figure 2C).

To establish a p63 gene regulatory network, we integrated the p63 cistrome with the cell line DEG datasets. This resulted in the identification of 1897 direct p63 targets and 3067 indirect targets (Figure 2D and Appendix A). For the direct targets, we found an enrichment of several oncogenic programs, including vascular endothelial growth factor (VEGF) signaling, focal adhesion, and TGF-β signaling in the p63^high^ B7E11 cells (Figure 2E). Conversely, both p63^low^ B7E3 and B4B8 cells were enriched for biological processes regulated by p63, such as cytodifferentiation, phosphoinositide-3 kinase pathway, EMT, and Notch1 signaling (Figure 2F).

### 3.3. Generating a p63-Driven mOSCC Transcriptomic Signature

To examine molecular processes regulated by p63 in B7E11 cells, we next performed knockdown studies. For these experiments, we used three stable, independent lentiviral shRNAs targeting *Trp63*. Western blotting confirmed that all three shRNAs (sh1, sh2, and sh3) reduced p63 expression, albeit at varying levels (Figure 3A). We performed RNA-seq on sh2 and sh3 cells, which affected the most robust p63 depletion. sh2 resulted in 100 DEGs, while sh3 resulted in 587 DEGs (Appendix A). We focused our analyses on sh3 as it produced the strongest knockdown of p63 at both the mRNA and protein levels.

Among the DEGs resulting from sh3-mediated p63 knockdown, 287 genes were upregulated and 300 were downregulated (Figure 3B,C). An analysis of enriched KEGG pathways using the DAVID bioinformatics database of these DEGs revealed cell cycle, epithelial–mesenchymal transition, apical junction, and the p53 signaling pathway among genes that were downregulated (Figure 3D). Conversely, upregulated DEGs were enriched in immune system-related pathways, including inflammatory responses and TNF signaling via NF-κB (Figure 3D). The upregulation of many cytokines and chemokines upon the loss of p63 is interesting because dysregulation of p63 has been associated with altered immune responses in SCC and other cancers [14,30,51,52,53].

To complement our p63 knockdown studies, we performed gain-of-function studies in the p63^low^ B7E3 cell line. For this purpose, we established a lentiviral-based doxycycline-inducible system to drive the expression of a FLAG epitope-tagged version of ΔNp63α. First, we confirmed that doxycycline (dox) induces ΔNp63α overexpression in a dose-dependent manner (Figure 4A). Next, we carried out RNA-seq experiments on cells in the presence and absence of dox. Differential gene expression analysis identified 147 DEGs (Figure 4B,C, Appendix A). To explore the pathways enriched in the 64 downregulated DEGs from the B7E3 p63-overexpressing cells, we performed GSEA using the hallmark gene set. The top enriched pathways in the downregulated genes were involved in IL2 signaling, hypoxia, and myogenesis (Figure 4D). Conversely, upregulated genes were enriched in p53 signaling, inflammatory responses, and EMT (Figure 4E). To validate the results from the RNA-seq results, we performed qRT-PCR for several of these genes, which confirmed the trends observed in both knockdown and overexpression experiments, indicating that p63 regulates the expression of these genes (Appendix A).

To identify a consensus set of genes that are most reliant on p63 expression, we combined the DEGs from both the loss- and gain-of-function experiments, revealing a set of 37 genes common to both datasets (Appendix A). To further identify genes that were directly regulated by p63, we incorporated our p63 ChIP-seq data, which revealed 22 genes directly bound by p63 in B7E11 cells (Appendix A). These analyses provided a p63-driven gene expression signature for 4NQO-induced mOSCC.

### 3.4. Evaluating the p63-Driven Gene Signature in an Independent 4NQO-Induced mOSCC Model

We explored if our p63-driven signature is also enriched in an independent RNA-seq dataset generated by Tang et al. [11] from tongue tumors in 6-week-old C57BL/6 mice treated with 100 μg/mL of 4NQO for 10 weeks. We reprocessed this data to perform DEG analysis between data from the 4NQO-treated and control tongues (mice treated with the vehicle) and identified 8171 DEGs (4014 upregulated and 4157 downregulated) at an adjusted *p*-value of ≤0.1 between the two conditions (Figure 5A and Appendix A). We focused on DEGs showing a log2 fold change of >1 to identify enriched pathways (Figure 5B). As expected, we found that *Trp63* was upregulated, in agreement with observations in other 4NQO studies (Appendix A). Overall, 2588 DEGs were downregulated upon tumor induction by 4NQO, and these were enriched for metabolic pathways, salivary secretion, and focal adhesion (Figure 5C). The enrichment of these pathways among downregulated genes matches that observed upon induction of p63 expression, suggesting that p63 plays a key role in this downregulation and that these pathways are key to 4NQO-induced tumorigenesis. By contrast, 2274 genes were upregulated in 4NQO-induced tumors (Appendix A). These upregulated DEGs displayed enrichment of pathways involved in cytokine–cytokine receptor interaction, chemokine signaling, cell cycle, and p53 signaling (Figure 5D). These pathways are similar to those identified for the p63-induced DEGs, highlighting the potentially important role of p63-signaling in 4NQO-induced tumors, particularly in regulating the tumor immune microenvironment.

We compared our p63-based signature with the list of DEGs from the tongue tumor dataset by Tang et al. [11] and identified 22 genes that were common to both (Figure 6A, Appendix A). These genes are involved in the p53 signaling pathway, cell adhesion pathways, and metabolic pathways (Figure 6A). Of the 22 genes, 14 are direct targets of p63 according to our high-confidence p63 target list (Figure 6A and Appendix A). Having ascertained a core signature for p63 signaling in murine OSCC, we turned our attention to human OSCC to determine if any of our identified targets may be conserved in the human disease context. In addition to human orthologs previously identified as playing key roles in OSCC, p63 signaling, and other cancers, namely, *FAT2*, *K14*, and *PERP*, we identified several novel regulators, including *Cotl1*, *Bcam*, *Adipor2*, and *Wnt7b*, (Figure 6A) [54,55,56,57,58]. Finally, to determine if the gene expression changes translate to differences in protein levels, we performed Western blotting on a subset of proteins encoded by DEGs. Overexpression of p63 in B7E3 cells resulted in an increase in the levels of Cotl1, Krt14, and Krt6 (Figure 6B), similar to what was seen in the RNA-seq data and demonstrating that the observed gene expression changes result in alterations to protein expression (Figure 6B).

### 3.5. scRNA-seq Analysis of 4NQO-Induced Mouse ESCC

In the absence of publicly available scRNA-seq data for 4NQO-generated OSCC, we turned our attention to esophageal squamous cell carcinoma (ESCC) [59]. Treatment of mouse esophageal tissues with 4NQO induces SCC in a manner that mimics the tumorigenic processes of ESCC in humans, similar to a 4NQO-induced mouse OSCC [59]. Therefore, we also employed a recently published dataset from scRNA-seq of a 4NQO-induced ESCC generated by Yao et al. [41] to explore the possible enrichment of our identified p63-based gene expression signature. First, we reanalyzed the ESCC scRNA-seq data generated by Yao et al. [41] and performed clustering analysis to identify nine overarching cell-type clusters (Figure 7A). We then assessed p63 expression across the tumor cell cluster and found a gradient of p63 expression across this cluster, with a clear population of cells with high expression and a population of cells with low expression (Figure 7B). Next, we performed DEG analysis on the identified p63^low^ cell population and p63^high^ population and found 480 genes, five of which overlapped our p63-based gene expression signature (Figure 7B, Appendix A). We also performed GSEA using the hallmark set of genes enriched in either the p63^low^ or p63^high^ cell population (Figure 7C). GSEA data from our p63 knockdown study markedly overlapped that from the p63^low^ population, sharing enrichment of KRAS signaling and interferon gamma response pathways (Figure 3E and Figure 7C). Likewise, in the p63^high^ cell population, the hallmark pathways overlap those in the cells with induced p63 overexpression, such as EMT, TNF signaling via NF-κB, and p53 signaling (Figure 4E and Figure 7D).

### 3.6. p63-Driven mOSCC Gene Signature Enriched in the Human TCGA Dataset

The identification of a p63 gene expression signature in mOSCC prompted us to examine its relevance in human tumors. To this end, we next explored the extent of cross-species conservation of the p63-driven signature in the TCGA-HNSCC datasets, keeping in mind the underlying differences in disease etiology and genetic complexity between these two cohorts. We compared the expression of 22 of the signature genes in the mouse dataset to that of the TCGA-HNSCC data. This analysis identified 16 genes that showed significant differences in expression between normal and tumor tissue (Appendix A). We expected six of these to be downregulated in tumors compared to expression in normal tissues; however, only two genes, *CES2* and *EMP1*, matched the pattern of expression identified in the mOSCC datasets utilized. (Appendix A). Similarly, 10 of the 15 genes upregulated in our signature, including *FAT2*, *COTL1*, and *KRT14*, were also upregulated in human tissues (Appendix A).

To determine the prognostic value of our signature, we performed Kaplan–Meier analyses for each of the 21 genes in our p63-driven signature (Appendix A). Only two genes were associated with differences in overall patient survival: higher expression of *BCAM* and *WNT7B* were associated with worse survival (Appendix A). A few other genes in our signature, such as *AMOTL2* and *ANXA3,* displayed trends toward a worse prognosis at higher levels, but these were not statistically significant (Appendix A). Overall, our analysis identified known and novel players in the p63 network that may be further investigated as drivers of the OSCC oncogenic program in both mice and humans.

### 3.7. p63 Affects mOSCC Cell Line Migration and Proliferation

p63 has been reported to regulate the proliferation and metastasis of human HNSCC cells, but its effect on mouse OSCC has not been explored [53,60,61,62]. Our previous pathway analysis revealed that B7E3 cells, which have undetectable levels of p63, are enriched for genes associated with processes such as cell adhesion and the cell cycle (compared to that of cells with high levels of p63). Thus, we examined if induced p63 expression influences the migratory and proliferative potential of B7E3 cells. We performed cell migration assays using a Transwell system and found that dox-induced p63 expression impaired the migration of cells compared to that of control cells (no dox) (Figure 8A), suggesting that p63 blocks migration. This finding is consistent with the downregulation of p63 expression in SCCs prior to cell migration and metastasis [25,63].

Next, to assess the effect of p63 expression on the proliferation of B7E3 cells in an anchorage-independent environment, we performed spheroid growth assays. B7E3 cells were grown either under control conditions with no dox or with 100 ng of dox to induce p63 overexpression. After 9 days, spheroids formed by p63-expressing B7E3 cells were significantly larger on average than spheroids grown from control cells (Figure 8B), confirming that p63 expression increases the proliferative ability of cells, including mOSCC cell lines.

### 3.8. COTL1 Is a Novel Target of p63 and Is Driven by p63 Expression

The enrichment of genes for several cell adhesion molecules in our combined p63 signature prompted us to look more closely at a novel p63 target, coactosin-like protein (Cotl1). COTL1 belongs to the ADF/cofilin family and likely binds to actin filaments to regulate the cytoskeleton [64]. COTL1 has been found to regulate the migration of mouse neocortical neurons and potentially plays a role in T-cell activation [65]. COTL1 also plays a role in the progression of cancer, promoting the proliferation of lung cancer and glioblastoma cells in vitro and in vivo [66,67]. *COTL1* mutations are one of the most prevalent subclonal mutations in human OSCC, with increased copy numbers in 10% of OSCC cases [68]. We assessed COTL1 expression in B7E3 cells by immunofluorescence staining using antibodies specific to p63 and COTL1 in B7E3 cells with and without dox-induced p63 expression. Although previous studies showed COTL1 expression to be exclusively cytoplasmic, we observed strong expression in nuclei (Figure 9). The nuclear staining for COTL1 overlapped that of DAPI in areas with low DNA density and remained in discreet aggregates within nuclei under all conditions, except in the case of actively dividing cells, in which it was more diffuse and surrounded the condensed DNA (Appendix A). The intensity of cytoplasmic staining of COTL1 matched that of p63, with stronger staining of cytoplasmic COTL1 with greater p63 induction (Figure 9). These findings are consistent with both our RNA-seq and Western blot results, showing that COTL1 expression is regulated by p63 expression.

To determine if these patterns of COTL1 and p63 expression in mOSCC cells occur in human OSCC tissues, we performed immunofluorescence staining of SCC tissues representing well-differentiated, moderately differentiated, and poorly differentiated SCC, because previous work has shown that p63 expression increases and expands as SCC becomes less differentiated [20]. Accordingly, we observed that p63 showed increased expression as well as expanded coverage of expression across the tissue from well-differentiated tissues to poorly differentiated tissues (Figure 10). We saw a similar pattern of expression for COTL1. Interestingly, COTL1 expression was tightly defined around endothelial tissue, suggesting the cytoplasmic COTL1 may play a role in maintaining basement membrane integrity (Figure 10).

Finally, we performed immunohistochemistry on a tissue microarray of normal and cancer human oral tissues, staining for p63, COTL1, and K14 (Figure 11). Similar to the results shown in Figure 10, we observed overall increased p63 staining in tumors compared to that in normal tissues. Notably, we found a stage-dependent increase in p63 expression, with advanced-stage tumors expressing the highest levels of p63 regardless of the anatomical site of the tumor (Figure 11). Similarly, the expression of K14, a known target of p63, was increased in cancer compared to that in normal tissue. Consistent with the results obtained from the previous immunofluorescence experiments, the pattern of COTL1 staining matched that of p63, with increased expression in cancer tissue (Figure 11). Taken together, these staining results confirm the gene expression-based findings of 4NQO-derived cell lines and tumors and further support the notion of an oncogenic network of p63 and its targets that operate in oral tumors.

## 4. Discussion

Tobacco smoking claims the lives of more than 6 million people every year worldwide and is one of the leading causes of cancer deaths in the United States [69,70]. The compound 4NQO is a precursor carcinogen that mimics some effects of tobacco, thus 4NQO-mediated chemical carcinogenesis models serve as valuable tools for the mechanistic exploration of tobacco-associated cancers, such as OSCC [71]. The molecular characterization of three representative 4NQO-induced OSCC cell lines through RNA-seq reported here provides a valuable resource and complements similar studies of the well-established MOC1 cell line [72,73]. One interesting observation from our studies is the difference in epithelial and mesenchymal gene expression patterns across the three mOSCC cell lines, in agreement with a prominent role for EMT in OSCC [49]. Specifically, we note that the B4B8 cell line displayed characteristics of a partial, or hybrid EMT state, as markers associated with both epithelial and mesenchymal states were highly expressed at the mRNA and protein levels. The hybrid EMT state of B4B8 cells may also predispose them to be particularly invasive, as shown previously [49], and might also be relevant in other contexts; for example, where it has been used as a murine HNSCC model to explore EGFR/ERBB-dependent growth. [74]

The p63-null B7E3 and the p63^high^ B7E11 cells offered us an excellent toolkit to study the role of the oncogenic ΔNp63 isoform in mOSCC and identify its transcriptional targets. Although previous studies have suggested p63 as a potential therapeutic target that is upregulated upon the deletion of the commonly mutated p16^INK4a^ locus in 4NQO-induced OSCC, the molecular and cellular processes that are regulated by p63 in this context have not been identified [12,24]. Our RNA-seq and ChIP-seq experiments in the p63^high^ B7E11 led us to the identification of genes regulated directly and indirectly by p63 and to a consensus p63-dependent core gene signature. By incorporating results from other independent and complementary datasets, we also examined the relevance of this mouse 4NQO-derived OSCC signature to human tumors [11,41]. The resultant signature includes genes well-known in OSCC and p63-dependent signaling networks, as well as several novel factors that shed light on the specific role of p63 in OSCC and even more broadly in SCC.

It is important to highlight that, unlike the genetically well-defined 4NQO models, human tumors result from multiple carcinogenic insults. Thus, one caveat is that cross-species comparisons of molecular signatures of OSCC are likely to reveal both similarities and differences. This is exemplified by *Bcam,* which encodes a cell adhesion protein that acts as a receptor for LAMA5, laminin that is a major component of the basement membrane [75]. BCAM is highly expressed in KRAS-mutant hepatic metastases from colorectal cancer, and inhibition of BCAM/LAMA5 interferes with the adhesion of colorectal cells to vascular endothelial cells, thereby reducing metastatic growth [76]. Intriguingly, while *BCAM* expression is increased in human HNSCC tumors, we found that *Bcam* expression is reduced in 4NQO-induced tumors, highlighting the differences between human and murine tumors. We observed higher *Bcam* expression in the p63^high^ cell population compared to the p63^low^ cells in the scRNA-seq generated by Yao et al. [41], further supporting the notion that this gene is regulated by p63. Since survival analyses suggest that BCAM expression does affect the outcome of human HNSCC, this might be a p63 target worthy of follow-up investigations. In addition, we also found significant enrichment of genes involved in the p53 signaling pathway, some of which have known interactions with p63, such as *Perp* and *Sfn* [77,78]. Interestingly, our analysis showed an upregulation of *Perp* in 4NQO-induced tongue tumors even though Perp is proapoptotic, suggesting a complicated balance of proliferative and apoptotic signaling [78]. Similarly, *Sfn* (Stratifin, also known as 14-3-3 protein sigma) was also increased in 4NQO-induced tumors, matching results from prior studies, which showed *Sfn* to be over-expressed in human HNSCC tumor samples in relation to non-cancerous head and neck tissues [79,80].

One of our notable findings was that the expression of several promigratory genes, including *Fat2* and *Cotl1*, paralleled that of p63 in 4NQO-induced tumors. p63 is known to induce the expression of human FAT2, as well as the mesenchymal gene Slug, to promote tumor invasion in breast cancer, whereas COTL1 increases the migratory ability of both breast and non-small cell lung cancer cells [56,81,82]. Interestingly, the p63 gene signature we identified also includes genes associated with invasion, such as *AMOTL2* that are paradoxically suppressed by p63. In humans, *AMOTL2* has conflicting roles in migration that seemingly depend on the cellular context, such that knockdown inhibits the migration of human umbilical vein endothelial cells but promotes EMT of mammary epithelial cells [83]. The role of *Amotl2* in mOSCC may more closely align with that in mammary epithelial cells, but that remains an area for further exploration. Similarly, we also observed that high p63 expression was associated with the downregulation of *Msln,* which encodes mesothelin, a protein that promotes EMT and invasion in breast, lung, and ovarian cancers [84,85,86]. A similar dichotomy of p63 function in cell migration and invasion has been reported before and has been attributed to an oscillatory expression of the ΔNp63 isoform that results in its tumor suppressive activities [25].

Finally, our exploration of the novel p63 target, COTL1 in the mouse cell lines and human HNSCC, suggests that it might have an oncogenic role that is conserved in both mouse and human disease. Notably, COTL1 and p63 have similar expression profiles in OSCC tissues, and higher expression of both was observed in more advanced disease stages, suggesting they play a role in the clinical outcome of OSCC patients. In support of this notion, COTL1 levels were amplified in up to 10% of patients with OSCC tumors in one study [68]. Previous immunofluorescence staining for COTL1 has shown diffuse localization in the cytoplasm; however, we observed both nuclear localization, as well as cytoplasmic staining of COTL1, which suggests an intriguing nuclear function for COTL1 [87]. The discrete aggregates of COTL1 in the nucleus, which seemingly disappear during cell division, hint at a possible role for COTL1 in cell division. Although the specific molecular function of COTL1 in OSCC requires additional studies, our results described here, combined with those previously reported, suggest COTL1 is involved in OSCC metastasis and patient survival. Additional support for this notion comes from a recent study identifying COTL1 as one of the 52 differentially expressed proteins in a proteomic-based comparison of metastatic and non-metastatic lymph nodes from HNSCC patients [88].

Overall, our studies reaffirm the previously identified role for p63 in modulating cellular processes that underlie tumor progression and metastasis, such as cell adhesion in OSCC, similar to what has been reported in other SCC and in epithelial development [16,89]. The p63-driven mOSCC signature and data generated from the 4NQO mOSCC cell lines set the stage for future studies into p63 function in OSCC and for the identification of actionable targets for future therapeutics. One major current limitation in this field is the lack of robust genomics datasets, in particular for 4NQO-derived tumors at different stages of tumor development and progression. Future transcriptomic and epigenomic studies addressing this shortcoming, particularly at single-cell resolutions, will offer valuable insights into the p63 gene regulatory and signaling networks and their role in the etiology of OSCC.

## 5. Conclusions

The dynamic regulation of p63 and its vast network of targets in both mouse 4NQO-derived cell lines and human tissues highlights its importance to OSCC. Our data re-emphasizes the crucial role of p63, specifically the ΔNp63 isoform in regulating networks of known and yet-to-be-discovered genes and a myriad of oncogenic processes, such as cell adhesion, migration, and metastasis that are important in OSCC. This is best illustrated by the identification of COTL1 as a novel factor regulated by p63 that can affect the clinical outcome of OSCC patients. How COTL1 functions in the cellular context and how its biological actions affect tumor cell behavior remains largely unknown. This remains the focus of mechanistic follow-up studies that are likely to be very informative. One limitation of our genomic studies is that they are based on cells grown in culture and thus likely do not represent an in vivo tumor. In this regard, syngeneic mouse models bearing tumors generated from the 4NQO-derived parental and modified cell lines can be extremely valuable to further validate our findings, especially the p63 gene signature. Taken together, the results from this study add to our understanding of the underlying molecular alterations associated with OSCC and provide potential targets for better diagnostics and therapy.

## Figures and Tables

**Figure 1 cancers-15-00446-f001:**
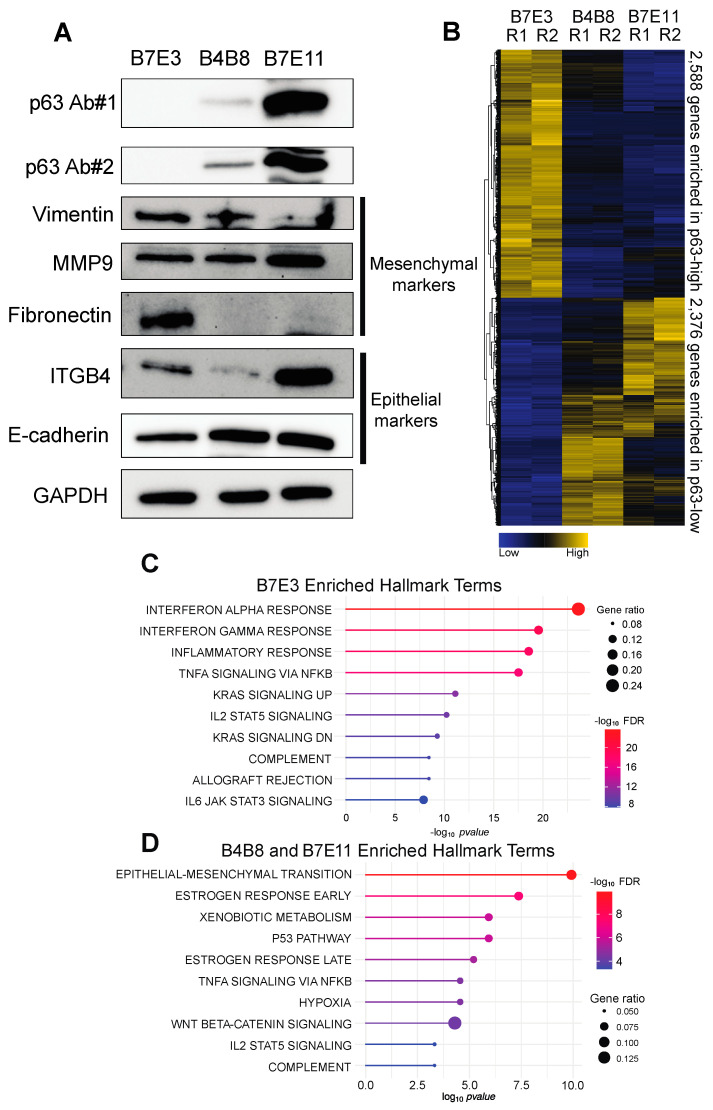
Profile of 4NQO-generated mOSCC cell lines. (**A**) Western blots for levels of p63 and EMT-associated factors in B7E3, B4B8, and B7E11 cell lines. (**B**) Heat map of the expression of genes consistently enriched in p63^high^ or p63^low^ cells. Enriched hallmark terms found in the p63^low^ B7E3 DEGs (**C**), the p63^high^ B4B8, and B7E11 DEGs (**D**).

**Figure 2 cancers-15-00446-f002:**
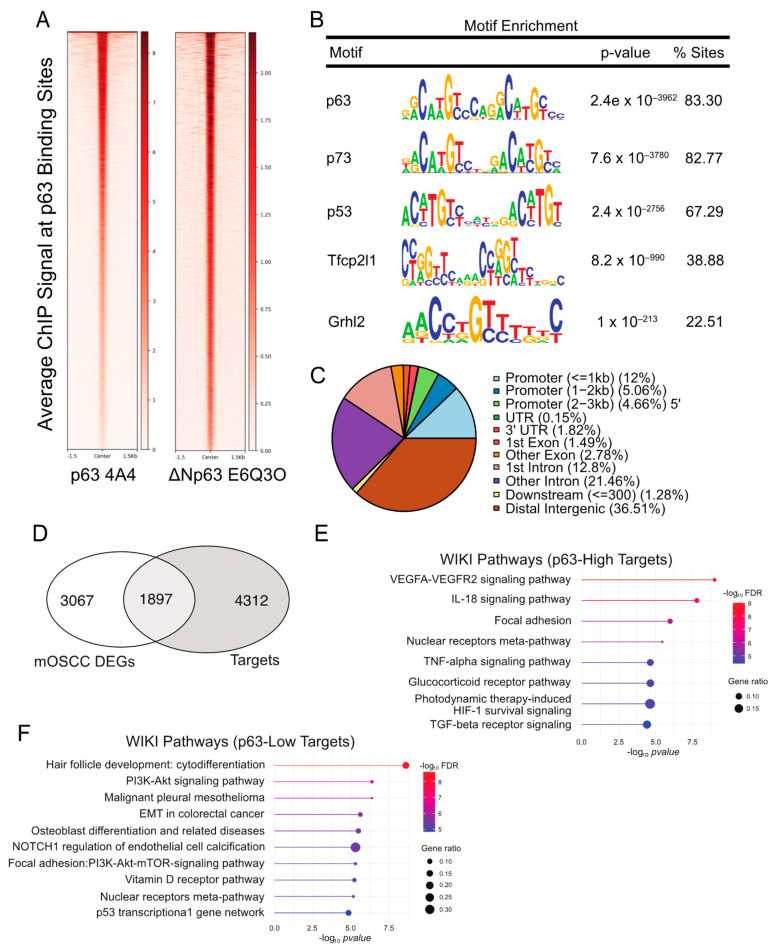
Identification of the p63 global targets in B7E11 cells by ChIP-seq. (**A**) Heat map of the average ChIP-seq signals from p63 binding sites for two p63 antibodies across the genome. (**B**) Top transcription factor motifs derived from CentriMo motif analysis on B7E11 consensus p63 ChIP peaks. (**C**) Distribution pattern of genomic features associated with p63 binding sites across the genome. (**D**) Integration of the identified p63 cistrome with the previously identified cell line-based DEG dataset. Enriched WikiPathways identified for p63^high^ (**E**) and p63^low^ (**F**) target genes.

**Figure 3 cancers-15-00446-f003:**
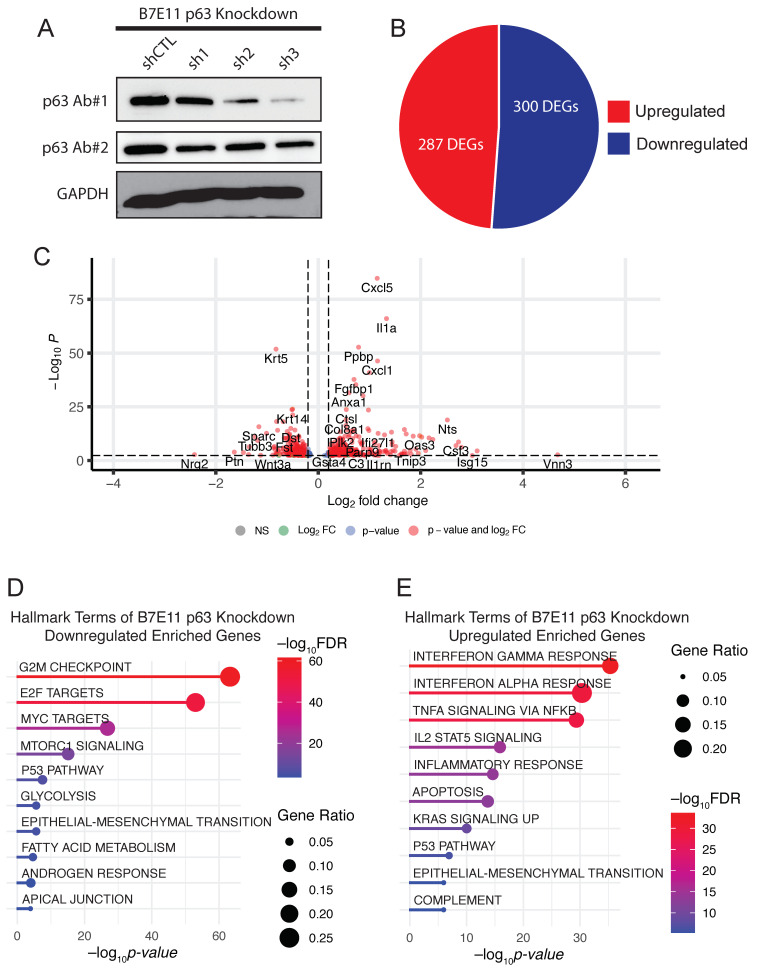
p63 knockdown in p63^high^ B7E11 cells. (**A**) Western blot analysis of p63 expression in B7E11 cells expressing either p63-targeting shRNAs or a nontargeting shRNA (shCTL). GAPDH was used as a loading control. A pie chart (**B**) and a volcano plot (**C**) of DEGs resulting from sh3 in B7E11 cells. Enriched hallmark terms in downregulated (**D**) and upregulated (**E**) DEGs in B7E11 cells with p63 knockdown.

**Figure 4 cancers-15-00446-f004:**
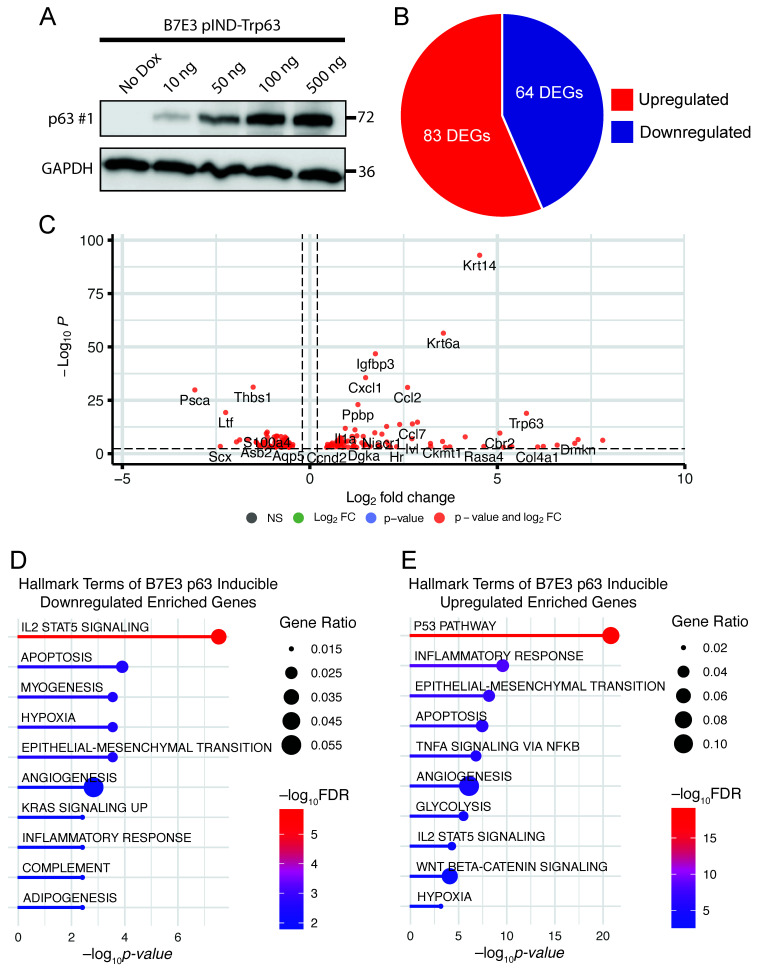
p63 overexpression in p63^low^ B7E3 cells. (**A**) Western blot analysis of p63 expression in B7E3 cells with dox-inducible p63 expression. GAPDH was used as a loading control. A pie chart (**B**) and a volcano plot (**C**) of DEGs resulting from induced p63 overexpression in B7E3 cells. Enriched hallmark terms in downregulated (**D**) and upregulated (**E**) DEGs in p63-overexpressing B7E3 cells.

**Figure 5 cancers-15-00446-f005:**
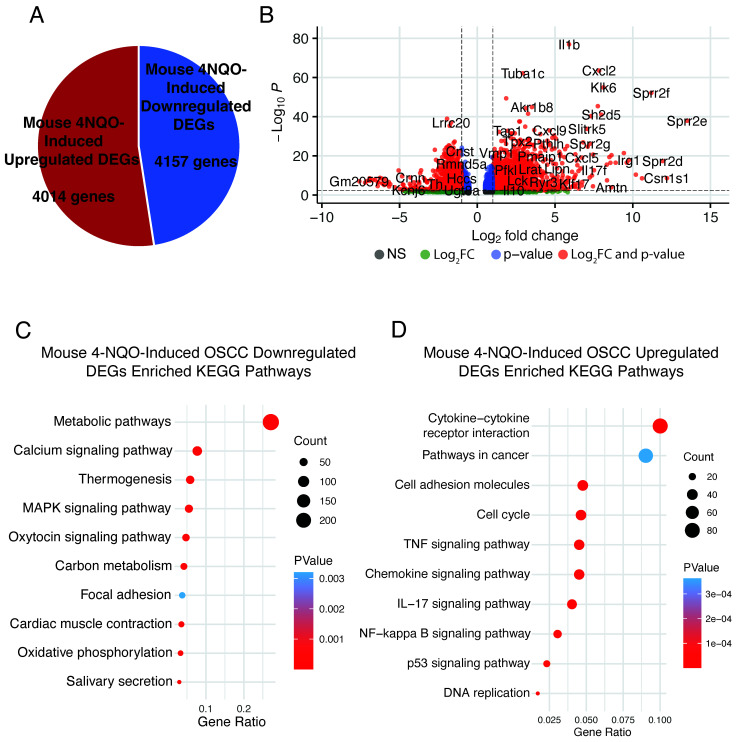
Investigating the p63-driven gene signature in an independent 4NQO-induced mOSCC model. A pie chart (**A**) and a volcano plot (**B**) of DEGs in the dataset by Tang et al. [11]. Enriched KEGG pathways in downregulated (**C**) and upregulated (**D**) DEGs from the dataset by Tang et al.

**Figure 6 cancers-15-00446-f006:**
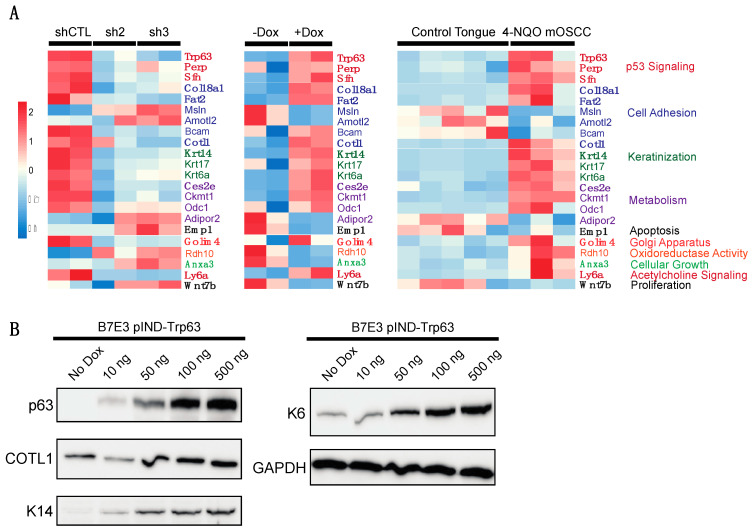
Combined mOSCC p63-based signature. (**A**) Heat map of the combined 22-gene signature in the B7E11 cells with p63 knockdown, B7E3 cells with induced p63 expression, and the Tang et al. [11] 4NQO mOSCC dataset. Genes in boldface are direct targets of p63, according to our high-confidence p63 target list. (**B**) Western blot analysis of the levels of proteins encoded by DEGs identified in our combined p63-based signature in B7E3 p63 overexpressing cells. GAPDH was used as a loading control.

**Figure 7 cancers-15-00446-f007:**
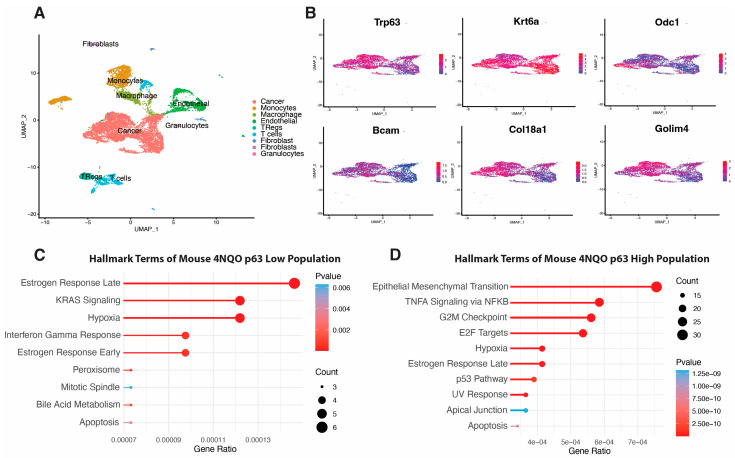
scRNA-seq analysis of 4NQO-induced mouse ESCC shows enrichment of our p63-based OSCC gene signature. (**A**) UMAP clustering of different cell types in mouse ESCC tumors. (**B**) UMAPs shows a zoomed-in cancer cell cluster highlighted from overall UMAP clustering. UMAP distribution of p63 expression across tumor cell clusters highlights p63^high^ and p63^low^ populations. DEG analysis of p63^high^ versus p63^low^ cells shows enrichment of several DEGs previously identified in the p63-based gene expression signature. UMAP projections display the expression of these factors across the tumor cell population. Enriched hallmark terms in DEGs enriched in p63^low^ (**C**) and p63^high^ (**D**) cells.

**Figure 8 cancers-15-00446-f008:**
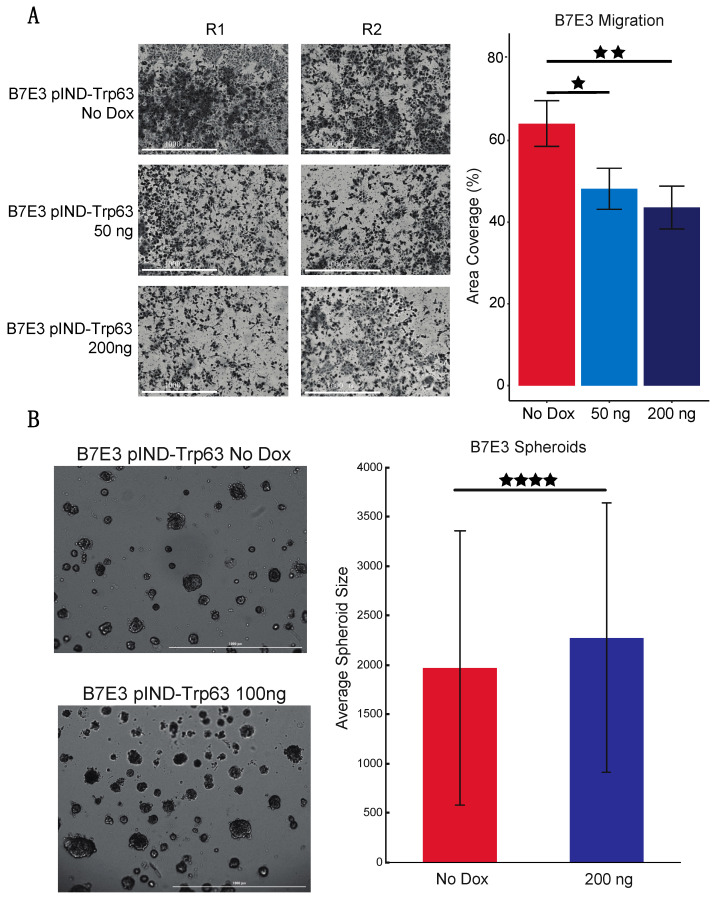
p63 regulates mOSCC cell line migration and proliferation. (**A**) Transwell migration assay with B7E3 cells with p63 expression induced by 50 or 200 ng dox. Cells that migrated through the Matrigel matrix by 12 h were then imaged and quantified by using ImageJ. (**B**) Spheroid assay with B7E3 cells with inducible p63 expression. Cells were treated with no dox or 100 ng of dox and allowed to form spheroids for 9 days. The sizes of the top 500 largest spheroids in both groups were then quantified using ImageJ. Values are presented as mean ± SD. Significance level was determined using two-tailed Student’s *t*-test for samples of equal variance. ^★^ *p* < 0.05, ^★★^ *p* < 0.01, and ^★★★★^ *p* < 0.001.

**Figure 9 cancers-15-00446-f009:**
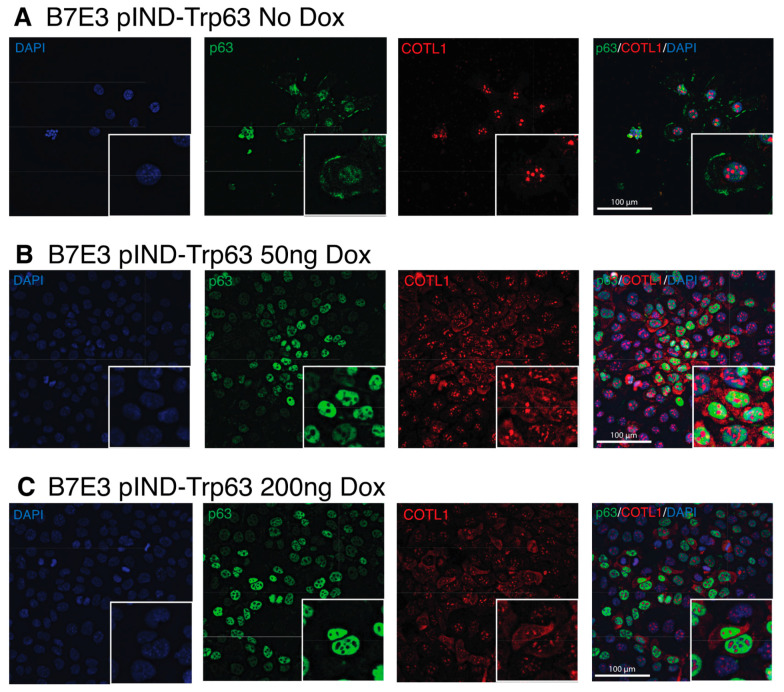
Co-staining of p63 and COTL1 in B7E3 cells with induced p63 expression. Immunofluorescence images of p63 (green), COTL1 (red), and DAPI (blue) in B7E3 cells treated with no dox (**A**), 50 ng dox (**B**), or 200 ng dox (**C**).

**Figure 10 cancers-15-00446-f010:**
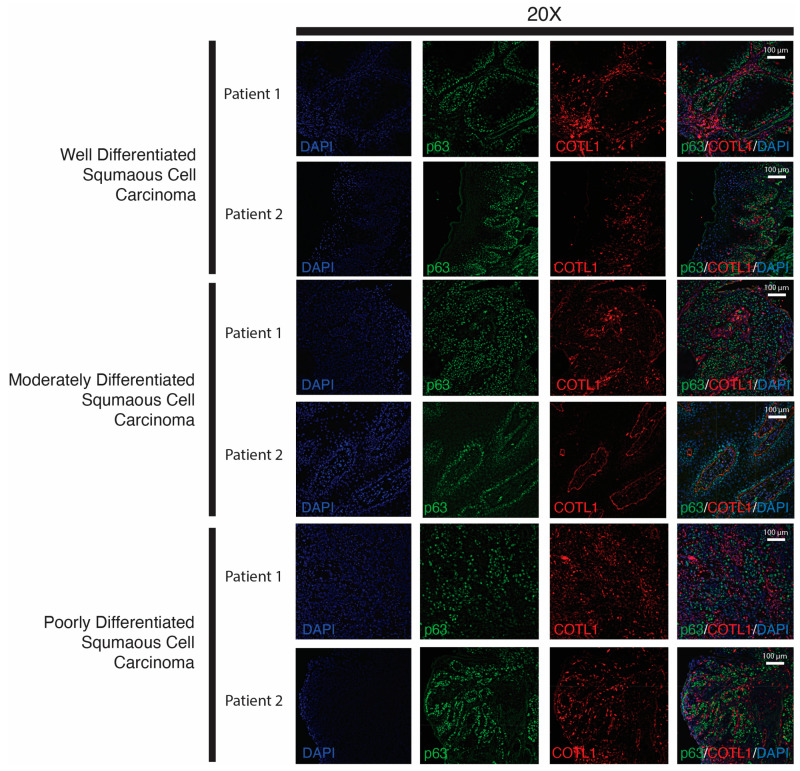
Co-immunofluorescence of p63 and COTL1 in human OSCC tissue. Two independent samples of well-differentiated, moderately differentiated, and poorly differentiated human OSCC tissues were co-stained for p63 (green) and COTL1 (red), and imaged at 20× magnification. DAPI (blue) was used as a nuclear stain.

**Figure 11 cancers-15-00446-f011:**
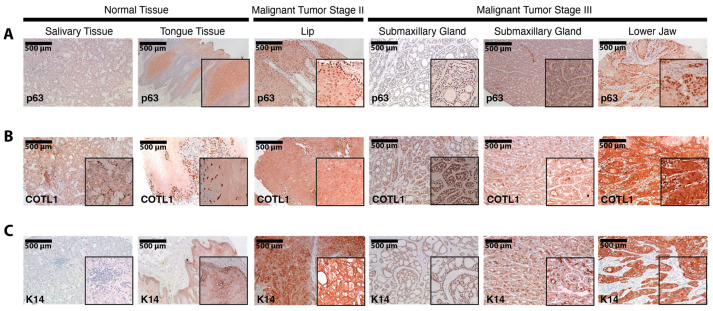
Immunohistochemical staining in HNSCC tumor microarray tissues. Staining of p63 (**A**), COTL1 (**B**), and K14 (**C**) across normal, malignant tumor stage II, and malignant tumor stage III tissues at 10× magnification.

## Data Availability

The datasets presented in this study are available at https://www.ncbi.nlm.nih.gov/geo/query/acc.cgi?acc=GSE217670 (accessed on 1 December 2022).

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
