# Peer review of "A Systemic and Integrated Analysis of p63-Driven Regulatory Networks in Mouse Oral Squamous Cell Carcinoma"

_cancers, 2023, doi:10.3390/cancers15020446_

Round 1
Reviewer 1 Report
The authors aimed to performed complementary loss- and gain-of-function experiments for p63 in mouse 4-nitroquinoline 1-oxide-transformed Oral squamous cell carcinoma (OSCC) cell lines, and to utilized RNA- and chromatin immunoprecipitation-sequencing to uncover the p63 oncogenic network. By combining their signature with the recent publicly available sequencing datasets, they generated a murine p63 signature to better understand the role of p63 in murine OSCC. Their analyses identified several potential biomarkers and conserved pathways that are likely to be also relevant to human OSCC and highlighted the dynamic role of p63 in regulating oncogenic processes such as migration and invasion.
The study covers some issues that have been overlooked in other similar topics. The structure of the manuscript appears adequate and well divided in the sections. Moreover, the study is easy to follow, but some issues should be improved. Some of the comments that would improve the overall quality of the study are:
a. Authors must pay attention to the technical terms acronyms they used in the text.
b. Better stated the aim of the study in the abstract and introduction section.
c. Conclusion Section: This paragraph required a general revision to eliminate redundant sentences and to add some "take-home message".
Author Response
- Authors must pay attention to the technical terms acronyms they used in the text.
We thank the reviewer for their rigorous evaluation and review. We have carefully checked over the technical language we used in this manuscript and made improvements where appropriate, particularly as it pertains to acronyms.
- Better stated the aim of the study in the abstract and introduction section.
We have made specific changes to the Abstract and Introduction to hopefully offer more clarity for the aims of our studies. Please see the revised manuscript where changes that were made are highlighted. We appreciate the advice.
- Conclusion Section: This paragraph required a general revision to eliminate redundant sentences and to add some "take-home message".
We thank the reviewer for their specific comment on the conclusion section. We have extensively revised the conclusion portion of the manuscript, which now provides our broad findings, limitations as well as more general “take-home messages”. These changed are highlighted in the revised manuscript and we hope that the Reviewer will find this to be satisfactory.
Reviewer 2 Report
In the present manuscript, Glathar and collaborators molecularly characterize different mouse 4NQO-derived OSCC sublines and identify a signature with potential prognostic value in OSCC that include novel p63 targets. They have directly related the specific signature with p63 expression and p63 driven oncogenic signaling.
The authors have done a lot of analysis and functional characterization to explore the role of p63 for the expression of different signatures and to get insight into the potential implications for cancer progression.
Additionally, the have complement their work in mouse models with human data what overall reinforces the work.
Overall the work is well structured and easy to follow. I would consider to revise the following aspects within the manuscript prior to publication:
- One important thing to consider is that within this work there is not a direct demonstration that p63 regulates the expression of Cotl1 and other genes contained in the signature that the authors described and these conclusions are based on undirect correlations. For this reason, I would suggest to change some of the sentences of the manuscript, e.g.:
“Results from our RNA-seq, ChIP-seq, and phenotypic experiments with mOSCC cell lines demonstrate that p63 regulates a broad range of pathways and targets including pathways affecting cell adhesion, migration, and metastasis, as well as the novel target, Cotl1.”
“To validate the results from the RNA-seq results, we performed qRT-PCR for several of these genes, which confirmed the trends observed in both indicating that p63 regulates the expression of these genes.”
“These findings are consistent with both our RNA-seq and Western blot results showing that COTL1 expression is regulated by p63 expression.”
I would suggest to indicate that the data suggest that p63 regulates COTL1 as well as other genes included within the signature.
- Related to Fig 1., It would be interesting to include an unsupervised principal component analysis (PCA) from the RNA seq data of B7E3, B7E11 and B4B8 cell lines to know if B4B8 and B7E11 present more similarities between them in an unbiased manner or they just share an specific set of genes that could be also related related to p63 expression.
- Related to Fig 2d, Fig. 2d, which cell line DEG data set correspond to the Venn diagram? Can the authors provide a pvalue for the overlap analysis?
- Related t Fig 7, can the authors include within the methods section a detailed explanation about how the “re-analysis” of ESCC sc-RNAseq data is performed?
- Also, there is a lack of product references within method section what given the case, would complicate the reproduction of the data by others.
- Finally, related to S4, it would be useful to include a prognostic value of a combination of genes or even a complete signature.
- Look over figure 7 legends, include Figs 8-11 scale bars.
Author Response
- One important thing to consider is that within this work there is not a direct demonstration that p63 regulates the expression of Cotl1 and other genes contained in the signature that the authors described and these conclusions are based on undirect correlations. For this reason, I would suggest to change some of the sentences of the manuscript. I would suggest to indicate that the data suggest that p63 regulates COTL1 as well as other genes included within the signature.
We agree with the Reviewer that we have primarily provided correlations for p63 targets specifically in relevance to patient or in vivo tumor data. For e.g., although it is clear that based on RNA-seq and ChIP-seq studies, COTL1 is likely a direct transcriptional target of p63, at least in the 4NQO cell lines, we have not taken the next step further to show a causal relationship. It is likely that not only other processes (such as post-transcriptional mechanisms), but also other transcription factors likely regulate the expression of COTL1. We have softened our choice of words to emphasize the “suggestive nature of the data” regarding the regulation of COTL1 as well as other genes included within the signature by p63.
- Related to Fig 1., It would be interesting to include an unsupervised principal component analysis (PCA) from the RNA seq data of B7E3, B7E11 and B4B8 cell lines to know if B4B8 and B7E11 present more similarities between them in an unbiased manner or they just share an specific set of genes that could be also related to p63 expression.
The reviewer has indeed brought up an interesting point and we agree that it will be useful to perform a more unbiased analysis of the transcriptome of the mouse 4NQO-induced cell lines in order to explore their similarities. Accordingly, we provide in the revised manuscript a PCA plot generated from an analysis of the top 2500 variable genes across each cell line (Figure S1). Our results show that the most variation lies between B7E3 and B7E11 cells, with B4B8 cells on a continuum path between the two as shown in Figure 1.
- Related to Fig 2d, Fig. 2d, which cell line DEG data set correspond to the Venn diagram? Can the authors provide a pvaluefor the overlap analysis?
We thank the reviewer for pointing this out, and apologize for not making this clear. The mOSCC DEGs used for the overlap analysis are the 4,964 genes reported in figure 1B as showing consistent changes in our DEG analysis.
The 1897 genes that were identified as overlapping in the analysis represent genes that are differentially expressed, and are predicted as targets based on proximity to a p63 binding site. This analysis, although binary in nature, was not analyzed as a probabilistic event, and therefore did not require multiple testing.
- Related t Fig 7, can the authors include within the methods section a detailed explanation about how the “re-analysis” of ESCC sc-RNAseq data is performed?
We thank the reviewer for bringing up this issue of method description which was lacking. We have updated our methods section to reflect a more detailed analysis of how we processed and analyzed all of the publicly available mouse bulk and single-cell RNA-seq data. Please see revised manuscript, specifically section 2.9, which details our updated methods.
- Also, there is a lack of product references within method section what given the case, would complicate the reproduction of the data by others.
We thank the reviewer for stressing the need to provide details of the products used in our studies to ensure the reproducibility of our data and have thus updated our methods section to provide more product details such as the company and product number. Please see revised methods section, which has changes tracked.
- Finally, related to S4, it would be useful to include a prognostic value of a combination of genes or even a complete signature.
As wisely recommended by the reviewer, we have been trying to do exactly that , i.e., to generate a prognostic value of a combination of p63 target genes or a complete p63-target gene signature. Interestingly, our findings so far are of mixed value and in some cases don’t meet statistical significance. For e.g., when we narrow it down to the 4 genes which showed greatest single gene survival differences (AMOTL2, ANXA3, BCAM, and WNT7B), most of the patients drop out and those that are left do not show a significant survival difference based on multi-gene Kaplan-Meier Estimator. But when we select just the two genes which show significant survival differences in the single gene Kaplan-Meier (BCAM, WNT7B) we do find one dataset (the TCGA PanCancer Atlas) in which it shows significant survival differences. We suspect these results are perhaps indicative of the higher genetic and molecular complexity of human OSCC cancers compared to the mouse 4NQO models. Adding to this conundrum is the dual role of p63 whose oscillatory expression levels might influence tumorigenesis both positively and negatively at different stages of cancer, such as metastasis. Given these results, we feel it will be prudent to pursue these aspects of p63 biology in future studies when more datasets become available. We hope the reviewer concurs.
- Look over figure 7 legends, include Figs 8-11 scale bars.
We thank the reviewer for their helpful suggestion and the scale bars have been added.